

# Performance impact of precision reduction in sparse linear systems solvers

Mawussi Zounon[1,2], Nicholas J. Higham[1], Craig Lucas[2] and Françoise Tisseur[1]

[1] School of Mathematics, University of Manchester, Manchester, United Kingdom
[2] The Numerical Algorithms Group, Manchester, Greater Manchester, United Kingdom

Corresponding author
Mawussi Zounon,
mawussi.zounon@manchester.ac.uk

## ABSTRACT

It is well established that reduced precision arithmetic can be exploited to accelerate the solution of dense linear systems. Typical examples are mixed precision algorithms that reduce the execution time and the energy consumption of parallel solvers for dense linear systems by factorizing a matrix at a precision lower than the working precision. Much less is known about the efficiency of reduced precision in parallel solvers for sparse linear systems, and existing work focuses on single core experiments. We evaluate the benefits of using single precision arithmetic in solving a double precision sparse linear system using multiple cores. We consider both direct methods and iterative methods and we focus on using single precision for the key components of LU factorization and matrix–vector products. Our results show that the anticipated speedup of 2 over a double precision LU factorization is obtained only for the very largest of our test problems. We point out two key factors underlying the poor speedup. First, we find that single precision sparse LU factorization is prone to a severe loss of performance due to the intrusion of subnormal numbers. We identify a mechanism that allows cascading fill-ins to generate subnormal numbers and show that automatically flushing subnormals to zero avoids the performance penalties. The second factor is the lack of parallelism in the analysis and reordering phases of the solvers and the absence of floating-point arithmetic in these phases. For iterative solvers, we find that for the majority of the matrices computing or applying incomplete factorization preconditioners in single precision provides at best modest performance benefits compared with the use of double precision. We also find that using single precision for the matrix–vector product kernels provides an average speedup of 1.5 over double precision kernels. In both cases some form of refinement is needed to raise the single precision results to double precision accuracy, which will reduce performance gains.

## INTRODUCTION

Ever since early versions of Fortran offered real and double precision data types, we have been able to choose between single and double precision floating-point arithmetics. Although single precision was no faster than double precision on most processors up to the early 2000s, on modern processors it executes twice as fast as double precision and

has the additional benefit of halving the data movement. As a result, single precision (as well as half precision) is starting to be used in applications such as weather and climate modelling (*Dawson et al., 2018*; *Váňa et al., 2017*) and seismic modeling (*Fabien-Ouellet, 2020*), where traditionally double precision was used. Mixed precision algorithms, which use some combination of half, single, double, and perhaps even quadruple precisions, are increasingly being developed and used in high performance computing (*Abdelfattah et al., 2021*).

In 2006, *Langou et al. (2006)* and *Buttari et al. (2007)*, drew the attention of the HPC community to the potential of mixed precision iterative refinement algorithms for solving dense linear systems with unprecedented efficiency. The underlying principle is to carry out the most expensive part of the computation, the LU factorization or Cholesky factorization, in single precision instead of double precision (the working precision) and then refine the initial computed solution using residuals computed in double precision. This contrasts with traditional iterative refinement, in which only a precision higher than the working precision is used. The resulting algorithms are now implemented in LAPACK (*Anderson et al., 1999*) (as DSGETRS, and DSPOTRS for general and symmetric positive definite problems, respectively), and are generally twice as fast as a full double precision solve for sufficiently well conditioned matrices.

A decade after the two-precision iterative refinement work by Buttari et al., Carson and Higham introduced a GMRES-based iterative refinement algorithm that uses up to three precisions for the solution of linear systems (*Carson & Higham, 2017*; *Carson & Higham, 2018*). This algorithm enabled Haidar et al. (*Haidar et al., 2018a*; *Haidar et al., 2020*; *Haidar et al., 2018b*) to successfully exploit the half-precision floating-point arithmetic units of NVIDIA tensor cores in the solution of linear systems. Compared with linear solvers using exclusively double precision, their implementation shows up to a $4\times$–$5\times$ speedup while still delivering double precision accuracy (*Haidar et al., 2020*; *Haidar et al., 2018b*). This algorithm is now implemented in the MAGMA library (*Agullo et al., 2009*; *Magma, 2021*) (routine magma_dhgesv_iteref_gpu) and in cuSOLVER, the NVIDIA library that provides LAPACK-like routines (routine cusolverDnDHgesv). Most recently, a five-precision form of GMRES-based iterative refinement has been proposed by *Amestoy et al. (2021)*, which provides extra flexibility in exploiting multiple precisions.

Mixed precision iterative refinement algorithms can be straightforwardly applied to parallel sparse direct solvers. But the variability of sparse matrix patterns and the complexity of sparse direct solvers make the estimation of the performance speedup difficult to predict. The primary aim of this work is to provide insight into the speedup to expect from mixed precision parallel sparse linear solvers. It is important to note that it is not our objective to design a new mixed precision algorithm, but rather we focus on analysing whether using single precision arithmetic in parallel sparse linear solvers has enough performance benefit to motivate mixed precision implementations.

After discussing existing work and the need for new studies we describe our experimental settings, including details of the sparse matrices and the hardware selected for our benchmark and analysis. We then introduce the issue of subnormal numbers appearing in single precision sparse LU factorization, explain how the subnormal numbers can

be generated, and propose different mitigation strategies. We present experimental performance results and show that by reducing the working precision from double precision to single precision for parallel sparse LU factorization, the expected speedup of 2 is only achieved for very large matrices. We provide a detailed performance profiling to explain the results and we present a similar analysis for iterative solvers by studying performance implications of precision reduction in sparse matrix–vector product and incomplete LU factorization preconditioner kernels.

## DISCUSSION OF EXISTING STUDIES

The performance benefits of mixed precision iterative refinement have been widely demonstrated for dense linear systems. The few such performance studies for sparse linear systems are summarized below, with an emphasis on the performance metrics reported.

### Mixed precision iterative refinement for sparse direct solvers

*Buttari et al. (2008)* studied the performance of mixed precision iterative refinement algorithms for sparse linear systems. They used Algorithm 1, in which the precision in which each line should be executed is shown at the end of the line, with FP32 denoting single precision and FP64 double precision. To implement Algorithm 1 they selected two existing sparse direct solvers: a multifrontal sparse direct solver MUMPS, by *Amestoy, Duff & L'Excellent (2000)* and a supernodal sparse direct solver SuperLU, by *Li & Demmel (2003)*. Multifrontal and supernodal methods are the two main variants of sparse direct methods; for a full description and a performance comparison see *Amestoy et al. (2001)*.

---

**Algorithm 1** Mixed-precision iterative refinement. Given a sparse matrix $A \in \mathbb{R}^{n \times n}$, and a vector $b \in \mathbb{R}^n$, this algorithm solves $Ax = b$ using a single precision sparse LU factorization of $A$ then refines $x$ to double precision accuracy.

| | | |
|---|---|---|
| 1: | Carry out the reordering and analysis for $A$. | |
| 2: | $LU \leftarrow \text{sparse\_lu(A)}$ | ▷ (FP32) |
| 3: | Solve $Ax = b$ using the LU factors. | ▷ (FP32) |
| 4: | **while** not converged **do** | |
| 5: | $\quad r \leftarrow b - Ax$ | ▷ (FP64) |
| 6: | $\quad$ Solve $Ad = r$ using the LU factors. | ▷ (FP32) |
| 7: | $\quad x \leftarrow x + d$ | ▷ (FP64) |
| 8: | **end while** | |

---

*Buttari et al. (2008)* showed that the version of SuperLU used in their study does not benefit from using low-precision arithmetic. Put differently, the time spent in matrix factorization, which is the most time-consuming part of the algorithm, is hardly reduced when single precision arithmetic is used in place of double precision. They concluded that a mixed precision iterative refinement based on SuperLU would be no faster than the standard double precision algorithm.

For MUMPS, their experimental results showed that the mixed precision version can be up to two times faster than the standard double precision MUMPS. While this result is consistent

with the performance observed for dense linear systems, there is an important difference to point out here: all the experimental results in *Buttari et al. (2008)* were obtained using a single core.

In 2010, *Hogg & Scott (2010)* designed a mixed precision iterative solver for the solution of sparse symmetric linear systems. The algorithm is similar to Algorithm 1, except they perform $LDL^T$ factorization instead of LU factorization and they also considered flexible GMRES (*Saad, 1993*) for the refinement process. Their experimental results show that the advantage of mixed precision is limited to very large problems, where the computation time can be reduced by up to a factor of two. But the results of this study are again based on single core benchmarks and also involve out-of-core techniques.

As these existing works are limited to a single core, further study is required to evaluate how the performance will be affected in fully-featured parallel sparse direct solvers using many cores. The main objective of using single precision arithmetic in sparse direct solvers is to reduce the time to solution. A safe way to improve performance without risking accuracy loss or inducing numerical stability is by exploiting the thread-level parallelism available in modern multicore processors. It is then sensible to first take advantage of core parallelism before using mixed precision algorithms for further performance enhancement. We aim to provide new insights into how far the exploitation of single precision arithmetic can advance the performance of parallel sparse solvers when computing a double precision accuracy solution.

## Mixed precision methods for iterative solvers

Here we summarize studies that use mixed precision arithmetic to improve the performance of iterative solvers. The existing works can be classified in three categories.

The first approach consists of using a single precision preconditioner or a few steps of a single precision iterative scheme as a preconditioner in a double precision iterative method. *Buttari et al. (2008)* have demonstrated the performance potential of this method using a collection of five sparse matrices, with a speedup ranging from 1.5x to 2.x. But the experiment has been performed on a single core using a diagonal preconditioner with an unvectorized sparse matrix–vector multiplication (SpMV) kernel.

The second approach, proposed in *Anzt et al. (2019)* and *Flegar et al. (2021)* uses low precision data storage whenever possible to accelerate data movement while performing all the computation in high precision. This concept is appealing, but hard to implement in practice as it requires an optimized data conversion routine and knowledge of key numerical properties of the matrices, such as the condition number. To illustrate this idea the authors of *Anzt et al. (2019)* designed a mixed precision block-Jacobi preconditioning method where the explicit inversion of the block diagonals is required.

The third category consists of studies that focus on designing a mixed precision SpMV kernel for iterative solvers. This approach has been implemented by *Ahmad, Sundar & Hall (2019)* by proposing a new sparse matrix format that stores selected entries of the input matrix in single precision and the remainder in double precision. Their algorithm accelerates data movement and computation with a small accuracy loss compared with double precision SpMV. Their implementation demonstrates up to $2\times$ speedup in the

best case, but hardly achieves any speedup on most of the matrices due to data format conversion overhead. A similar approach has been implemented by *Grigoraş et al. (2016)* with a better speedup for FPGA architectures.

Our contribution is to assess from a practical point of view the benefit of using single precision arithmetic in iterative solvers for a double precision accuracy solution, by evaluating optimized vendor kernels used in applications.

## EXPERIMENTAL SETUP

The experimental results are reported using the Intel dual-socket Skylake with 40 cores and the NVIDIA V100 GPU. We have also performed experiments using the AMD dual-socket EPYC Naples system with 64 cores and the NVIDIA P100 GPU; and we obtained similar results. We note that the arithmetic properties of the NVIDIA GPUs are investigated in *Fasi et al. (2021)*. The sparse matrices selected for the benchmark are from various scientific and engineering applications and are summarized in Table 1. The Intel Skylake node has 50 GB of main memory, and consequently sparse matrices whose factors require more than 50 GB storage are not included. The matrices are divided in two groups. The first 21 matrices are from the medium size group with 700,000 to 5,000,000 nonzero elements. It takes a few seconds on average to factorize these matrices. The second group contains larger matrices with 7,000,000 to 64,000,000 nonzeros and it takes on average a few minutes to factorize most of the matrices in this group. For each matrix, the largest absolute value $\max_{i,j}|a_{ij}|$ and the smallest nonzero absolute value $\min_{i,j}\{|a_{ij}| : a_{ij} \neq 0\}$ of the elements are reported in Table 1. For medium size matrices, an estimate for the 1-norm condition number, $\kappa_1(A) = \|A^{-1}\|_1 \|A\|_1$, computed using the MATLAB `condest` routine, is also provided.

For each experiment, we consider the average time over 10 executions and we clear the L1 and L2 caches between consecutive runs.

## APPEARANCE OF SUBNORMAL NUMBERS IN SINGLE PRECISION SPARSE LU AND MITIGATION TECHNIQUES

From Table 1, one can observe that the entries of the matrices fit in the range of single precision arithmetic, which from Table 2 we see comprises numbers of modulus roughly between $10^{-45}$ and $10^{38}$. There is no risk of underflow or overflow in converting these matrices to single precision format. However, the smallest absolute value of matrix `ASIC_320ks`, $1.26 \times 10^{-39}$, is a subnormal number in single precision. A subnormal floating-point number is a nonzero number with magnitude less than the absolute value of the smallest normalized number (*Higham, 2002*, Chap. 2), (*Muller et al., 2018*, Chap. 2). Floating-point operations on subnormals can be very slow, because they often require extra clock cycles, which introduces a high overhead.

**Table 1  Selected matrices from the SuiteSparse Matrix Collection (*Davis, 2021*; *Davis & Hu, 2011*). The first 21 matrices are of medium size and each can be factorized in a few seconds. Matrices 22 to 36 are larger and require more time and memory to solve.**

|  | Matrix | Size | nnz | $\kappa_1(A)$ | $\max_{i,j}|a_{ij}|$ | $\min_{i,j}\{|a_{ij}| : a_{ij} \neq 0\}$ |
|---|---|---|---|---|---|---|
| 1 | 2cubes_sphere | 101,492 | 1,647,264 | 2.93e+09 | 2.52e+10 | 6.68e−15 |
| 2 | ASIC_320ks | 321,67 | 1,316,085 | 5.06e+22 | 1.00e+06 | 1.26e−39 |
| 3 | Baumann | 112,211 | 748,331 | 1.368+09 | 1.29e+04 | 5.00e−02 |
| 4 | cfd2 | 123,440 | 3,085,406 | 3.66e+06 | 1.00e+00 | 6.66e−09 |
| 5 | crashbasis | 160,000 | 1,750,416 | 1.78e+03 | 4.08e+02 | 6.42e−11 |
| 6 | ct20stif | 52,329 | 2,600,295 | 2.22e+14 | 8.86e+11 | 3.02e−34 |
| 7 | dc1 | 116,835 | 861,071 | 1.01e+10 | 5.67e+4 | 3.00e−12 |
| 8 | Dubcova3 | 146,689 | 3,636,643 | 1.14e+04 | 2.66e+00 | 8.47e−22 |
| 9 | ecology2 | 999,999 | 4,995,991 | 6.66e+07 | 4.00e+01 | 1.00e+01 |
| 10 | FEM_3D_thermal2 | 147,900 | 3,489,300 | 1.66e+03 | 2.92e−01 | 1.16e−05 |
| 11 | G2_circuit | 150,102 | 726,674 | 1.97e+07 | 2.22e+04 | 3.27e−01 |
| 12 | Goodwin_095 | 100,037 | 3,226,066 | 3.43e+07 | 1.00e+00 | 1.41e−21 |
| 13 | matrix-new_3 | 125,329 | 893,984 | 3.47e+22 | 1.00e+00 | 1.27e−21 |
| 14 | offshore | 259,789 | 4,242,673 | 2.32e+13 | 7.47e+14 | 7.19e−21 |
| 15 | para-10 | 155,924 | 2,094,873 | 8.13e+18 | 6.44e+11 | 2.26e−20 |
| 16 | parabolic_fem | 525,825 | 3,674,625 | 2.11e+05 | 4.00e−01 | 3.18e−07 |
| 17 | ss1 | 205,282 | 845,089 | 1.29e+01 | 1.00e+00 | 1.06e−11 |
| 18 | stomach | 213,360 | 3,021,648 | 8.01e+1 | 1.38e+00 | 1.47e−09 |
| 19 | thermomech_TK | 102,158 | 711,558 | 1.62e+20 | 1.96e+02 | 4.83e−03 |
| 20 | tmt_unsym | 917,825 | 4,584,801 | 2.26e+09 | 4.00e+00 | 1.00e+00 |
| 21 | xenon2 | 157,464 | 3,866,688 | 1.76e+05 | 3.17e+28 | 5.43e+23 |
| 22 | af_shell10 | 1,508,065 | 52,259,885 | | 5.72e+05 | 1.00e−06 |
| 23 | af_shell2 | 504,855 | 17,588,875 | | 1.51e+06 | 4.55e−13 |
| 24 | atmosmodd | 1,270,432 | 8,814,880 | | 2.22e+04 | 3.19e+03 |
| 25 | atmosmodl | 1,489,752 | 10,319,760 | | 7.80e+04 | 3.96e+04 |
| 26 | cage13 | 445,315 | 7,479,343 | | 9.31e−01 | 1.15e−02 |
| 27 | CurlCurl_2 | 806,529 | 8,921,789 | | 4.42e+10 | 8.84e+06 |
| 28 | dielFilterV2real | 1,157,456 | 48,538,952 | | 6.14e+01 | 3.25e−13 |
| 29 | Geo_1438 | 1,437,960 | 60,236,322 | | 6.69e+12 | 4.75e−07 |
| 20 | Hook_1498 | 1,498,023 | 59,374,451 | | 1.58e+05 | 5.17e−26 |
| 31 | ML_Laplace | 377,002 | 27,689,972 | | 1.22e+07 | 1.24e−09 |
| 32 | nlpkkt80 | 1,062,400 | 28,192,672 | | 2.00e+02 | 4.08e−01 |
| 33 | Serena | 1,391,349 | 64,131,971 | | 5.51e+13 | 2.19e−01 |
| 34 | Si87H76 | 240,369 | 10,661,631 | | 1.83e+01 | 2.57e−13 |
| 35 | StocF-1465 | 1,465,137 | 21,005,389 | | 3.10e+11 | 9.57e−09 |
| 36 | Transport | 1,602,111 | 23,487,281 | | 1.00e+00 | 1.62e−12 |

The risk of underflow, overflow or generating subnormal numbers during the conversion from higher precision to lower precision can be reduced using scaling techniques proposed by *Higham, Pranesh & Zounon (2019)*. However, even if matrices have been safely converted

**Table 2 Parameters for IEEE single and double precision point arithmetic.** $x_{\min,s}$ is the smallest nonzero subnormal number and $x_{\min}$ and $x_{\max}$ are the smallest and largest normalized floating-point numbers.

|  | $x_{\mathbf{min},s}$ | $x_{\mathbf{min}}$ | $x_{\mathbf{max}}$ | Unit roundoff |
|---|---|---|---|---|
| FP32 | $1.4 \times 10^{-45}$ | $1.2 \times 10^{-38}$ | $3.4 \times 10^{38}$ | $6.0 \times 10^{-8}$ |
| FP64 | $4.9 \times 10^{-324}$ | $2.2 \times 10^{-308}$ | $1.8 \times 10^{308}$ | $1.1 \times 10^{-16}$ |

from double to normalized single precision numbers, subnormal numbers may still be generated during the computation. We first suspected this behavior in our benchmark when some single precision computations took significantly more time than the corresponding double precision computations. For example, the sparse direct solver MUMPS computed the double precision $LU$ decomposition of the matrix Baumann (#3 in Table 1) in 1.6251 seconds, while the single precision factorization took 3.586 seconds. Instead of being two times faster than the double precision computation, the single precision computation is two times slower. A further analysis reveals that the smallest magnitude entries of the single precision factors $L$ and $U$ are of the order of $10^{-88}$, which is a subnormal number in single precision but a normalized number in double precision. The appearance of subnormal numbers in the single precision factors may be surprising, since the absolute values of the entries of this matrix range from $5 \times 10^{-2}$ to $1.29 \times 10^4$, which appears to be innocuous for single precision.

This phenomenon of LU factorization generating subnormal numbers does not appear to have been observed before. How can it happen? The elements at the $(k+1)$st stage of Gaussian elimination are generated from the formula

$$a_{ij}^{(k+1)} = a_{ij}^{(k)} - m_{ik}a_{kj}^{(k)}, \quad m_{ik} = \frac{a_{ik}^{(k)}}{a_{kk}^{(k)}},$$

where $m_{ik}$ is a multiplier. If $A$ is a dense matrix of normalized floating-point numbers with norm of order 1, it is extremely unlikely that any of the $a_{ij}^{(k)}$ will become subnormal. However, for sparse matrices we can identify a mechanism whereby fill-in cascades down a column and small multipliers combine multiplicatively. Consider the upper Hessenberg matrix

$$A = \begin{bmatrix} d_1 & 0 & \cdots & \cdots & 0 & 1 \\ -a_1 & d_2 & 0 & \cdots & 0 & 0 \\ & -a_2 & d_3 & 0 & \cdots & \vdots \\ & & -a_3 & d_4 & \ddots & \vdots \\ & & & \ddots & \ddots & 0 \\ & & & & -a_{n-1} & d_n \end{bmatrix}.$$

LU factorization without row or column permutations produces the LU factorization

$$
LU \equiv
\begin{bmatrix}
1 & & & & \\
-\dfrac{a_1}{d_1} & 1 & & & \\
& -\dfrac{a_2}{d_2} & \ddots & & \\
& & \ddots & 1 & \\
& & & -\dfrac{a_{n-1}}{d_{n-1}} & 1
\end{bmatrix}
\begin{bmatrix}
d_1 & 0 & \dots & \dots & 0 & 1 \\
& d_2 & 0 & \dots & 0 & \dfrac{a_1}{d_1} \\
& & d_3 & 0 & \dots & \dfrac{a_1 a_2}{d_1 d_2} \\
& & & d_4 & \ddots & \vdots \\
& & & & \ddots & \dfrac{a_1 a_2 \dots a_{n-2}}{d_1 d_2 \dots d_{n-2}} \\
& & & & & d_n + \dfrac{a_1 a_2 \dots a_{n-1}}{d_1 d_2 \dots d_{n-2}}
\end{bmatrix}.
$$

The elements $-a_i/d_i$ on the subdiagonal of $L$ are multipliers. The problem is in the last column of $U$. If $|a_i/d_i| < 1$ for all $i$ then $|u_{in}|$ will decrease monotonically with $i$, and if $|a_i/d_i| \ll 1$ for many $i$ then $|u_{in}|$ will eventually become subnormal as $i$ increases. This can happen because of large $d_i$ or small $a_i$. As illustrated in this example, subnormal numbers are mainly generated in the fill-in process, with zero entries gradually replaced with subnormal numbers. Consequently, sparse matrix reordering algorithms for fill-in reduction can naturally help decrease the appearance of subnormal numbers, but unless fill-in is fully eliminated, different mitigation techniques are required to prevent performance drop.

The performance loss caused by arithmetic on subnormal numbers is often mitigated by two options: Flush to Zero (FTZ) and Denormals[1] Are Zero (DAZ). With the FTZ option, when an operation results in a subnormal output, zero is returned instead, while with the DAZ option any subnormal input is replaced with zero. For the sake of simplicity we will refer to both options as FTZ in the rest of this paper. It may be possible to enable the FTZ option using compiler flags. For example this is automatically activated by Intel's C and Fortran compilers whenever the optimization level is set higher than `-O0`. However, we have used the GNU Compiler Collection (GCC) in this study, and the only option to flush subnormals to zero is via the `-fast-math` option. But the `-fast-math` flag is dangerous as it also disables checking for `NaN`s and `+-Inf`s and does not maintain IEEE arithmetic compatibility, so it can result in incorrect output for programs that depend on an IEEE-compliant implementation (https://gcc.gnu.org/onlinedocs/gcc/Optimize-Options.html). As a safe alternative to the `-fast-math` flag, we use the x86 assembly code; see the listing in Fig. 1. Calling SetFTZ() before the factorization routines guarantees flushing subnormals to zero without compromising the numerical robustness of the software. Once the SetFTZ() routine is called at the beginning of a program, it is effective during the whole execution, unless it is explicitly deactivated by calling another $\times 86$ assembly code not listed in this paper.

[1] Subnormal numbers are also referred to as denormal numbers.

## SINGLE PRECISION SPEEDUP OVER DOUBLE PRECISION FOR SPARSE LU FACTORIZATION

The main performance gain of mixed precision iterative refinement algorithms comes from using low precision arithmetic to factorize the coefficient matrix associated with the linear system. The factorization stage dominates the cost of the algorithm, assuming that

```
1   void  SetFTZ(void_zou)
2     {
3     asm("stmxcsr -0x4(%rsp)\n\t"     /* store MXCSR register on stack  */
4         "orl $0x8040,-0x4(%rsp)\n\t"  /* set bits 15(FTZ) and 7(DAZ)    */
5         "ldmxcsr -0x4(%rsp)");        /* load MXCSR register from stack */
6     }
```

**Figure 1** **x86 assembly code for flushing subnormals to zero, while maintaining IEEE arithmetic compatibility.**

the refinement converges quickly. We therefore focus on the speedup achieved during the matrix factorization step to evaluate the potential of low-precision arithmetic for solving sparse linear systems. For each problem from Table 1, we report the speedup achieved during the factorization, and we use a threshold of $1.5\times$ to decide whether low precision is beneficial. Note that in the case of dense linear systems, the factorization step speedup is usually close to $2\times$.

In addition to SuperLU and MUMPS, we have added PARDISO (*Schenk et al., 2001*), which is available in the Intel Math Kernel Library (MKL), to the set of sparse direct solvers for the benchmarks. PARDISO combines left- and right-looking level 3 BLAS supernodal algorithms for better parallelism. The solvers also include the multithreaded version of SuperLU, called SuperLU_MT *Li (2005)*. We will refer to both packages as SuperLU unless there is ambiguity. We also considered adding UMFPACK *Davis (2004)*, but this package does not have support for single precision.

For each sparse direct solver, we report the factorization speedup for both sequential and parallel runs. Even though the Intel Skylake has 40 cores, we report parallel results with 10 cores as for most of the experiments the performance stagnates and sometimes declines beyond 10 cores. To stress the performance penalty induced by subnormals in the single precision computations, the results with and without FTZ are reported.

The experimental results with serial PARDISO are summarized in Fig. 2. For each matrix two bars are shown, which give the speedup for LU factorization with and without FTZ. Without FTZ, up to 15 matrices out of 36 show a speedup below 1. In other words, single precision decreases the performance for 42% of the problems compared with double precision. This anomaly is corrected by flushing subnormals to zero. By comparing the results with FTZ with results without FTZ, we see that more than half of the problems generated subnormals during the single precision computation. As for the performance benefit of using single precision for the matrix factorization, half of the matrices show a speedup above the $1.5\times$ threshold. The matrices that did not exceed $1.5\times$ speedup are predominately of medium size. The parallel results in Fig. 3 show that with 10 cores the proportion of problems that reach $1.5\times$ speedup drops from 50% to 30%. The problems that still reach $1.5\times$ speedup with 10 cores are exclusively from the large matrices and represent 65% of them.

The results for serial MUMPS are summarized in Fig. 4. The matrices that suffered performance degradation due to subnormals in the PARDISO experiments exhibit similar behavior with MUMPS. Similarly, half of the matrices did not reach the threshold of $1.5\times$,

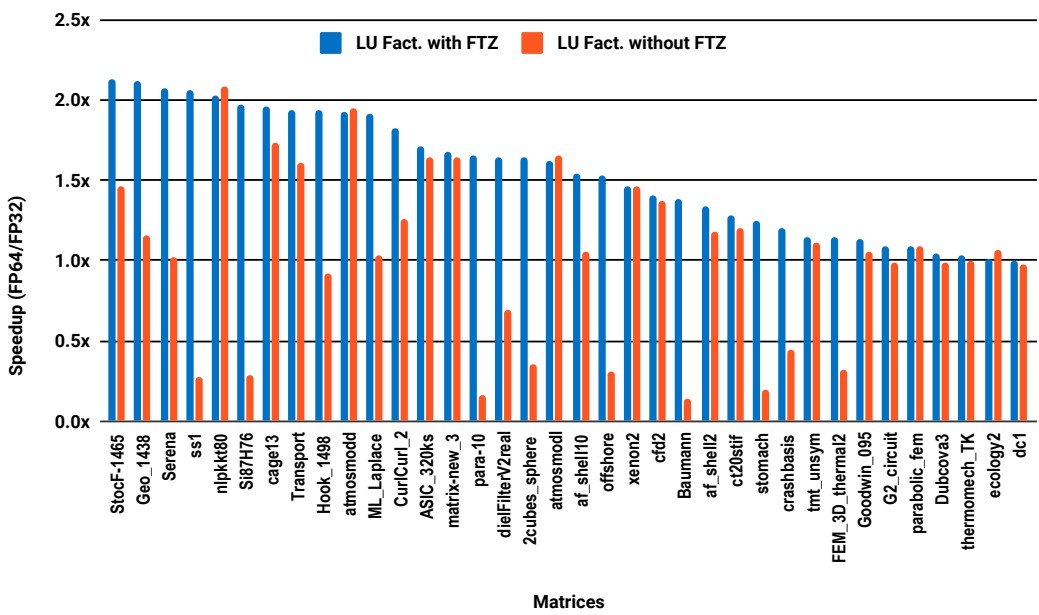

**Figure 2** Single precision speedup over double precision for sparse LU factorization using PARDISO on a single Intel Skylake core.

and the matrices beyond 1.5× are mainly the large ones. The parallel results in Fig. 5 are less attractive as only five matrices deliver a speedup beyond 1.5×. These matrices are from the large size group.

Unlike PARDISO and MUMPS, the multithreaded SuperLU ran out of memory for 15 problems out of the 36, predominantly the large size ones. Results are reported for only the 21 remaining matrices. The serial results in Fig. 6 show that only 33% of the 21 problems, successfully solved exceed 1.5x speedup, against 24% for the parallel results in Fig. 7.

These results show that mixed precision iterative refinement may only be beneficial for large sparse matrices. However, a large matrix size and higher density are not enough to predict the speedup, as matrix dielFilterV2real is much larger and denser than cage13 but its speedup is lower than cage13's speedup in all the experiments. We note the contrast with dense linear systems, where a 2× speedup is often achieved even for matrices of size as small as $200 \times 200$.

## ANALYSIS OF RESULTS FOR SPARSE LU FACTORIZATION

Apart from the unforeseen high occurrence of subnormal numbers in single precision sparse LU factorization, two other unexpected observations require further explanation. These are the poor speedup of the matrices from the medium size group, and the fact that many matrices show better speedup in single core experiments than with parallel execution. This section aims to address these questions.

Sparse direct solvers employ more elaborate algorithms than dense solvers. Given a sparse linear system to solve, the rows and the columns of the sparse matrix are first reordered to reduce the number of nonzero elements in the factors, or such that the matrix

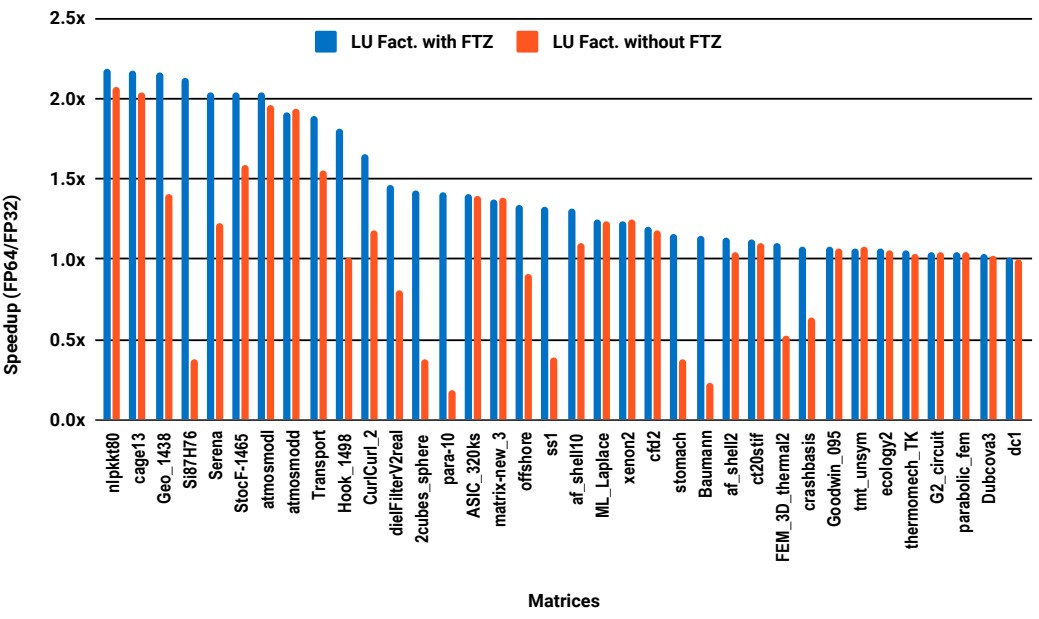

**Figure 3** Single precision speedup over double precision for sparse LU factorization using `PARDISO` on 10 Intel Skylake cores.

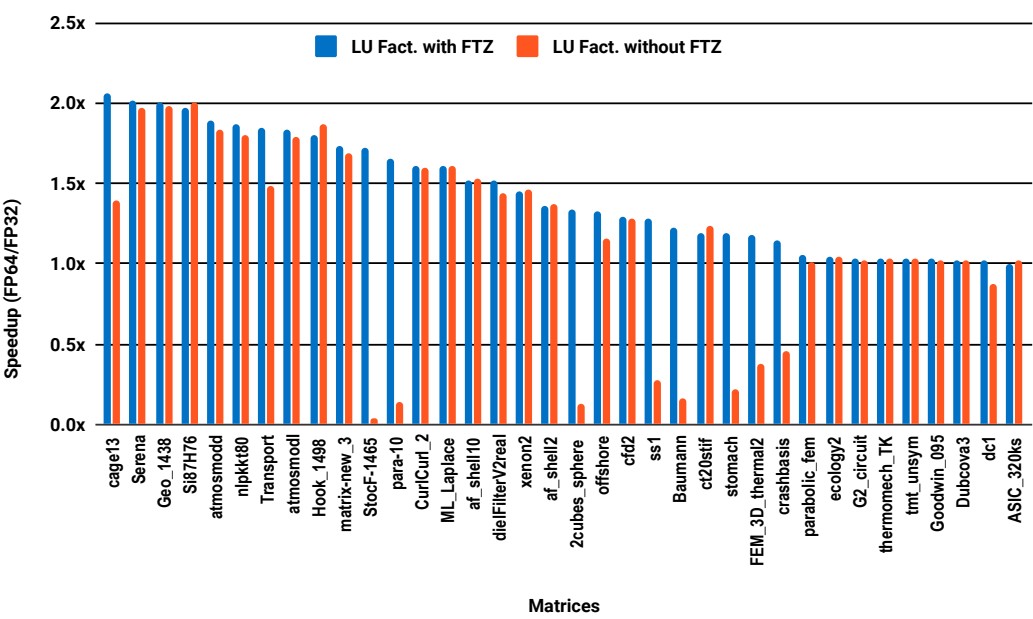

**Figure 4** Single precision speedup over double precision for sparse LU factorization using `MUMPS` on a single Intel Skylake core.

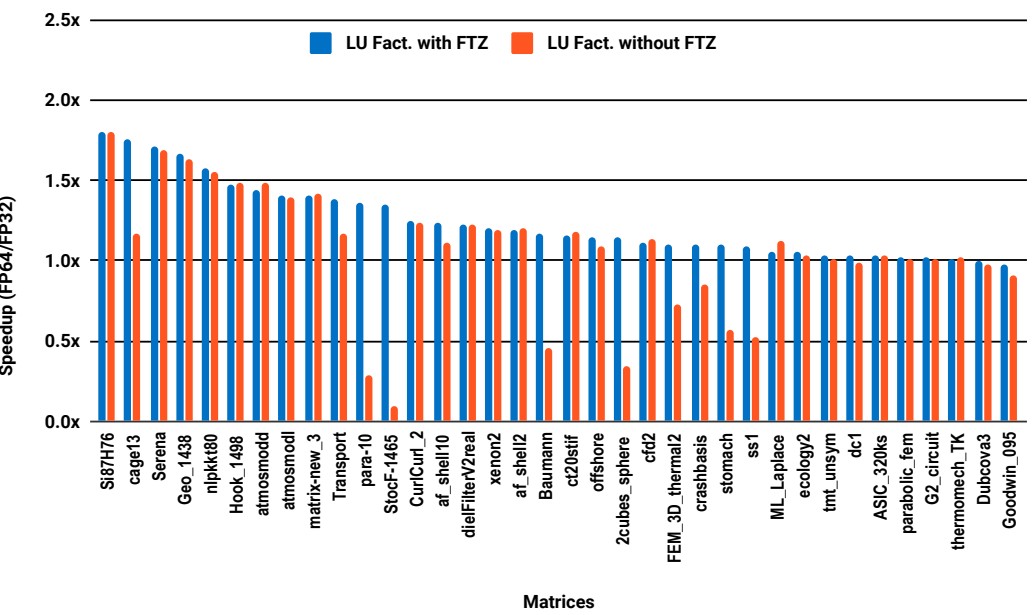

**Figure 5** Single precision speedup over double precision for sparse LU factorization using MUMPS on 10 Intel Skylake cores.

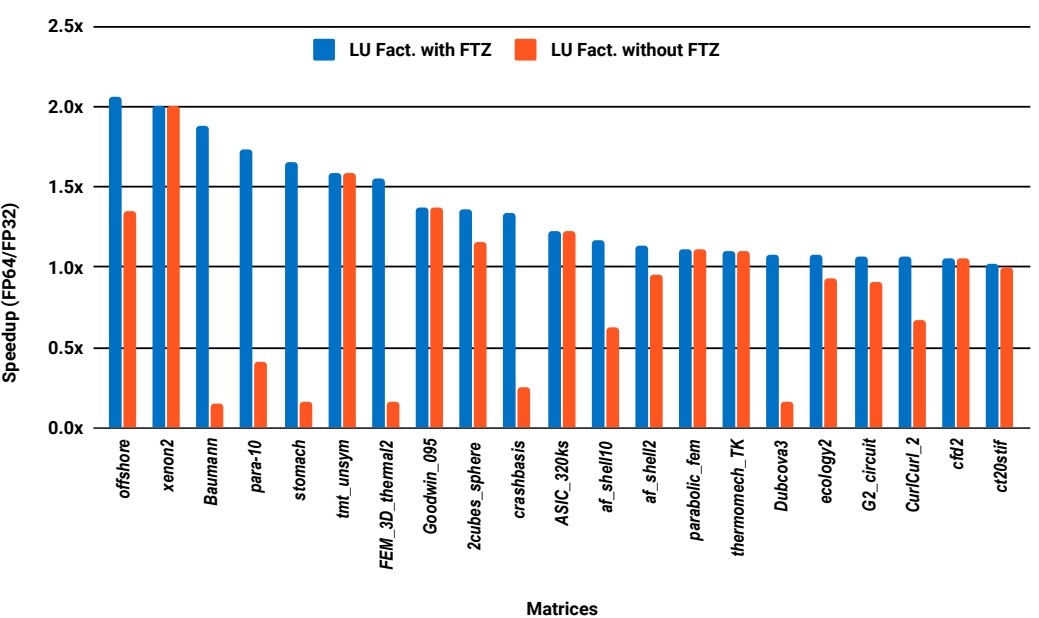

**Figure 6** Single precision speedup over double precision of sparse LU factorization using SuperLU on a single Intel Skylake core. SuperLU ran out of memory for 15 problems.

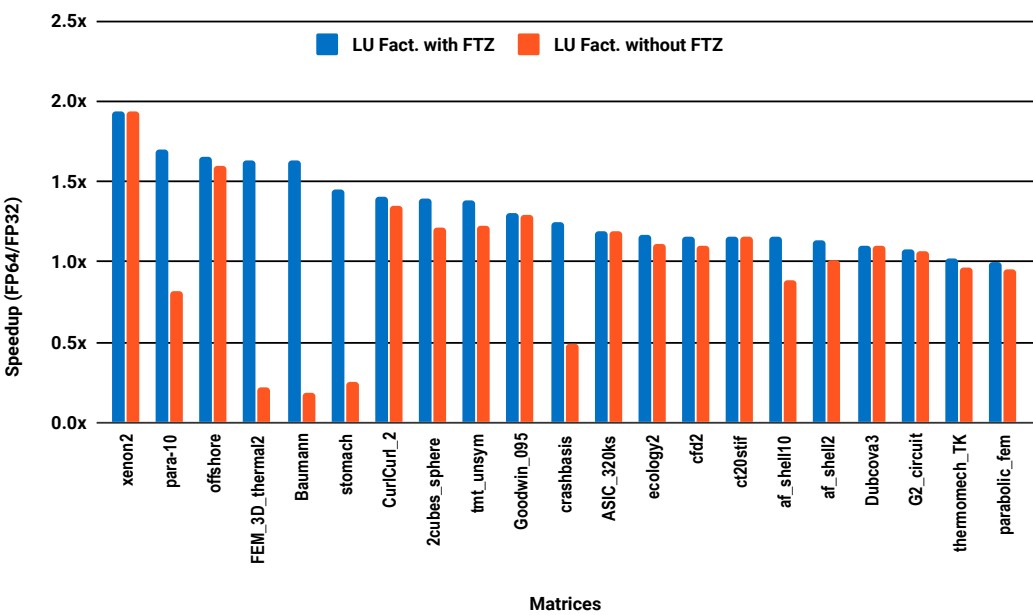

**Figure 7** **Single precision speedup over double precision for sparse LU factorization using SuperLU on 10 Intel Skylake cores.** SuperLU ran out of memory for 15 problems.

has dense clusters to take advantage of BLAS 3 kernels. This pre-processing step is called reordering, and it is critical for the overall performance and the memory consumption. After the ordering, the resulting matrix is analyzed to determine the nonzero structures of the factors and allocate the required memory accordingly. This step is called symbolic factorization. It is followed by the numerical factorization step that computes the LU factors, and finally the solve step.

The reordering and the analysis steps do not involve floating-point arithmetic. Therefore, they do not benefit from lowering the arithmetic precision. If the reordering and the analysis represent 50% of the overall factorization time, for example, then using single precision instead of double will only reduce the overall time by a quarter in the best case. This explains the poor speedup on average size matrices compared with the large size group. This is illustrated in Fig. 8 where one can observe that the majority of average size matrices spend more than 25% of the overall time in the reordering and analysis steps. The matrices for which the reordering and analysis time is negligible are the ones that reach up to 2× speedup with single precision. In general, the matrix sparsity pattern and the effectiveness of the reordering algorithms will impact the speedup observed. For example, for some small or moderate size matrices with a complex sparsity pattern, some reordering algorithms may suffer a large amount of fill-in, causing the cost of the numerical factorization to dominate and leading to a significant benefit from using single precision. Further analysis of how the speedup depends on the matrix characteristics and the fill-in rate observed during the symbolic and reordering steps is outside the scope of this work.

The second issue, the decrease of speedup in parallel experiments compared with single core executions, is due to the lack of parallelism in the reordering and analysis steps. For

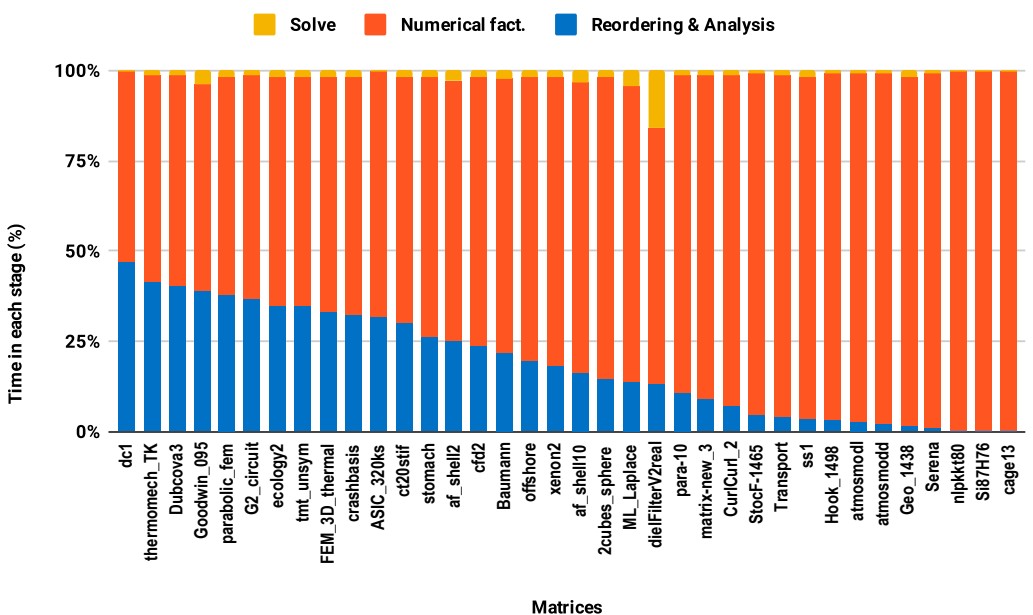

**Figure 8** **Time spent by double precision sequential PARDISO LU in each step on a single Intel Skylake core.** The bars are sorted by decreasing time associated with the reordering and analysis step.

example in this work, all the sparse solvers except PARDISO use sequential reordering and analysis algorithms on shared memory multicore architectures. PARDISO provides the parallel version of the nested dissection algorithm for reordering, but compared with the sequential version, it reduces the reordering time only by a factor of 2 while the numerical factorization time decreases significantly, by up to a factor of 8 using 10 cores. Consequently, by increasing the number of cores, the proportion of time spent in reordering and analysis steps increases as illustrated in Fig. 9. One can observe that in the parallel experiment, half of the matrices spent more than 50% of the overall factorization time in reordering and analysis, which explains the limited acceleration from lowering the precision.

## SINGLE PRECISION SPEEDUP OVER DOUBLE PRECISION FOR SPARSE ITERATIVE SOLVERS

The performance of an iterative solver depends not only on the algorithm implemented but also on the eigenvalue distribution and condition number of the matrix, the choice of preconditioner, and the accuracy targeted. It is therefore hard to make general statements about how mixed precision techniques will affect the performance of an iterative solver. Therefore, in this section, we focus instead on analyzing the impact of low precision in SpMv kernels and preconditioners, as they are the building blocks of iterative solvers.

The results in Fig. 10 illustrate the speedup from using single precision incomplete LU factorization (ILU0) from the cuSPARSE (https://docs.nvidia.com/cuda/cusparse) library on an NVIDIA V100 GPU. The cuSPARSE library provides an optimized implementation of

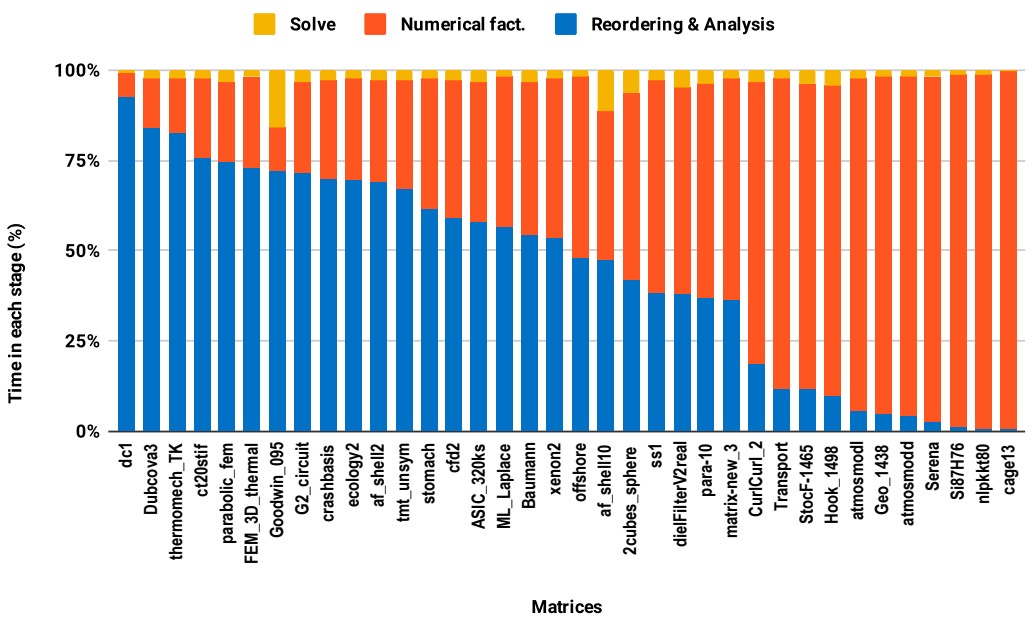

**Figure 9** **Time spent by double precision parallel PARDISO LU in each step on 10 Intel Skylake cores.** The bars are sorted by decreasing time associated with the reordering and analysis step.

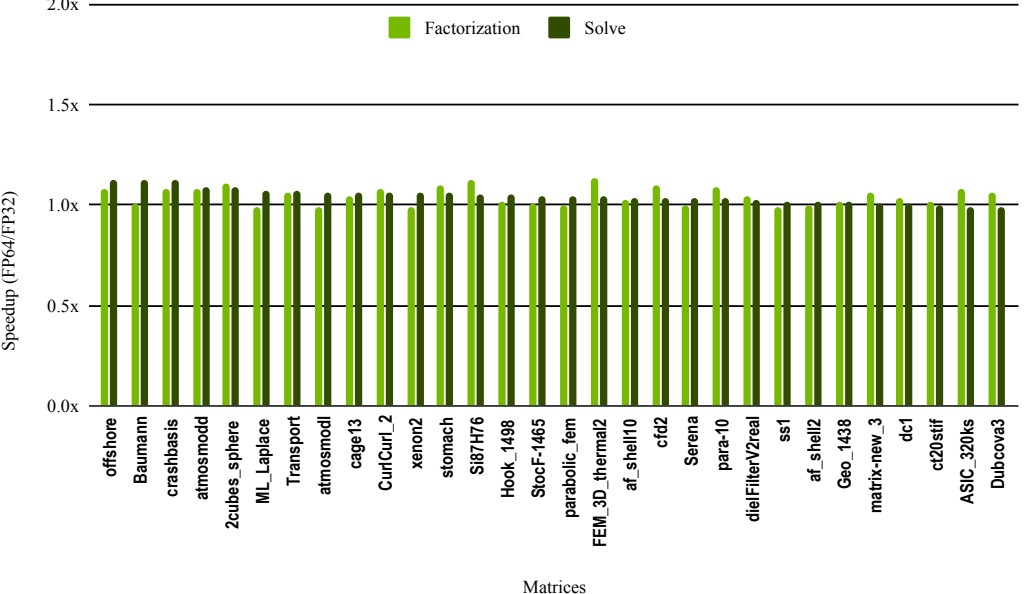

**Figure 10** **Speedup of single precision versus double precision for sparse incomplete LU factorization (ILU0) using cuSPARSE on NVIDIA V100 GPU.**

a set of sparse linear algebra routines for NVIDIA GPUs. For the sake of readability, the matrices are sorted in a decreasing order of the solve step speedup.

The most critical part of the preconditioner application is the forward and backward solve, because it is executed at each iteration and can easily become the most time consuming

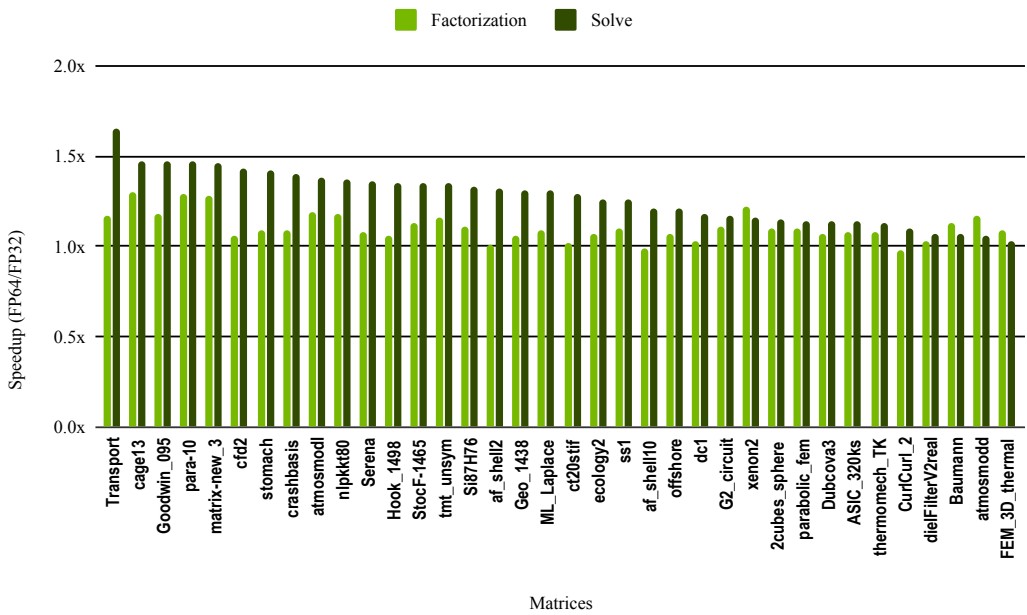

**Figure 11** **Speedup of single precision versus double precision for sparse incomplete LU factorization (ILU) using SuperLU on Intel Skylake.** The SuperLU ILU implementation is serial but it has been compiled against a multithreaded MKL BLAS and run with 10 cores.

part of iterative solvers. The dark green bars in Fig. 10 represent the speedup of the single precision ILU0 preconditioner application. The performance shows that lowering the precision in the preconditioner application did not enhance the performance. The same is true for the incomplete factorization itself, so there is no benefit to using single precision in place of double precision. The results from SuperLU ILU in Fig. 11 show a better speedup for the solve step compared with the results from cuSPARSE ILU0. However, the speedup is still under the threshold of 1.5× speedup, except for one matrix (Transport). For the incomplete LU factorization step itself, the performance gain from using single precision is insignificant. As the factorization step is more time-consuming than the solve steps, the overall speedup of the preconditioner computation and application remains very small and does not seem to present enough potential to accelerate parallel iterative solvers. Note that from the libraries evaluated in this work only SuperLU and cuSPARSE provide incomplete LU factorization implementation.

To evaluate how low precision can accelerate SpMV kernels, we have considered the compressed row storage (CSR) format, as it is widely used in applications. In the CSR format, a double precision sparse matrix with $nnz$ nonzero elements requires approximately $12nnz$ bytes for the storage (each nonzero element requires 8 bytes for its value and 4 bytes for its column index). In single precision the matrix will occupy approximately $8nnz$ bytes of memory. As SpMV kernels are memory bandwidth-bound, the use of single precision will only provide a 1.5x ($12nnz$ divided by $8nnz$) speedup in theory. Note that, for simplicity we have ignored the $4n$ bytes for row indices, where $n$ is the number of rows, and the extra memory for left- and right-hand side vectors. The results in Fig. 12 for the optimized

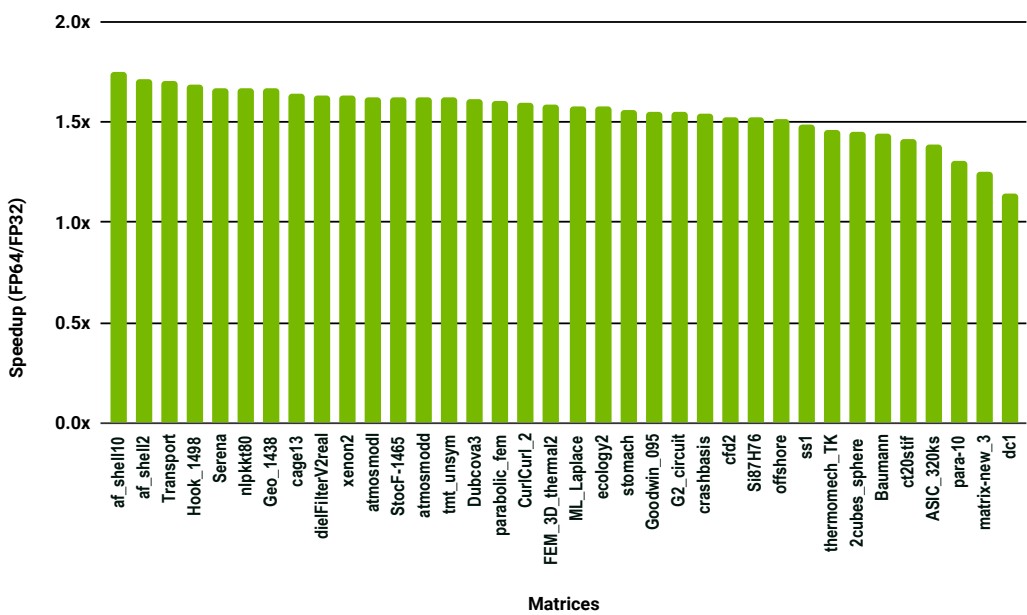

**Figure 12** Speedup of single precision *versus* double precision for SpMV using cuSPARSE on NVIDIA V100 GPU.

cuSPARSE SpMV on the NVIDIA V100 GPU show that the speedup is oscillating around 1.5x. Similarly, the benchmark of the MKL SpMV in Fig. 13 shows that the single precision kernel has approximately $1.5\times$ speedup over the double precision kernel.

This study shows that computing or applying the ILU preconditioner in single precision usually offers at best a modest speedup over double precision. Taking advantage of efficient single precision SpMV kernels typically gives a 1.5 speedup. However, in both cases the results will have at best single precision accuracy, so some form of refinement to double precision will be necessary, which will reduce the speedups.

## CONCLUSION

The benefits of using single precision arithmetic to accelerate compute intensive operations when solving double precision dense linear systems are well documented in the HPC community. Much less is known about the speedup to expect when using single precision arithmetic in parallel algorithms for double precision sparse linear systems, and existing work focuses on single core experiments. In this work, we have assessed the benefit of using single precision arithmetic in solving double precision sparse linear systems on multicore architectures. We have evaluated two classes of algorithms: iterative refinement based on single precision LU factorization and iterative methods using single precision for the matrix–vector product kernels or preconditioning.

Our first finding is that a limiting factor in the performance of single precision sparse LU factorization is the generation of subnormal numbers, which occurs for the majority of our test matrices. We have identified a mechanism whereby fill-in can cascade down a column, creating and then propagating subnormal numbers with it. We have demonstrated

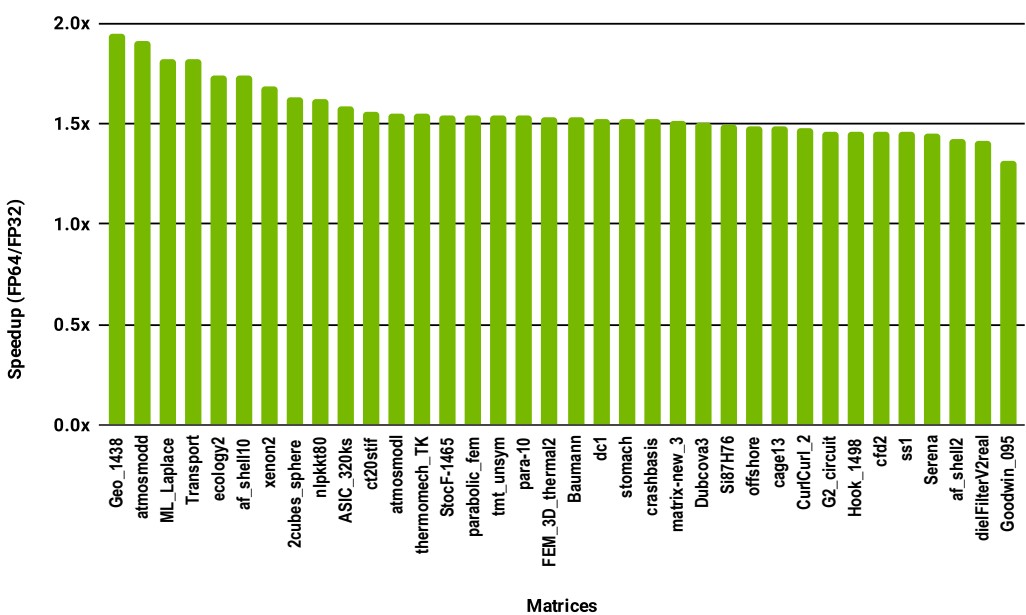

**Figure 13** **Speedup of single precision *versus* double precision for SpMV using MKL on 10 Intel Skylake cores.**

the severe performance drop that can result and have shown how flushing subnormals to zero can mitigate it.

Our second finding is that the anticipated speedup of 2 from using single precision arithmetic is obtained only for the very largest of our test problems, where the analysis and reordering time is negligible compared with numerical factorization time.

Our last finding concerns iterative solvers. Our results show that the performance gain in computing or applying incomplete factorization preconditioners in single precision is typically much less than a factor 1.5, but we have observed a speedup of around 1.5 by evaluating matrix–vector product kernels in single precision. In future work, we will explore new approaches to integrate efficiently single precision matrix–vector product kernels and single precision preconditioners in double precision iterative solvers without accuracy loss.

Finally, we note that half precision arithmetic is of growing interest, because of the further benefits it brings through faster arithmetic and reduced data movement. For dense systems, GMRES-based iterative refinement(discussed in the introduction) successfully exploits a half precision LU factorization to deliver double precision accuracy in the solution. We are not aware of any half precision implementations of sparse LU factorization but if and when they become available we hope to extend our investigation to them.

### Funding

This work was supported by Innovate UK under grant number KTP011064, by the Engineering and Physical Sciences Research Council under grant number EP/P020720/1, and by the Royal Society. The funders had no role in study design, data collection and analysis, decision to publish, or preparation of the manuscript.

### Grant Disclosures

The following grant information was disclosed by the authors:
Innovate UK: KTP011064.
The Engineering and Physical Sciences Research Council: EP/P020720/1.
The Royal Society.

### Competing Interests

Nicholas J. Higham is an Academic Editor for PeerJ. Mawussi Zounon is employed by the University of Manchester and The Numerical Algorithms Group. Craig Lucas is employed by the Numerical Algorithms Group. We state that there is no competing interest resulting from the fact that Mawussi Zounon and Craig Lucas are working for a non-academic institution.

### Author Contributions

- Mawussi Zounon conceived and designed the experiments, performed the experiments, analyzed the data, performed the computation work, prepared figures and/or tables, authored or reviewed drafts of the paper, and approved the final draft.
- Nicholas J. Higham conceived and designed the experiments, analyzed the data, prepared figures and/or tables, authored or reviewed drafts of the paper, and approved the final draft.
- Craig Lucas and Françoise Tisseur analyzed the data, prepared figures and/or tables, authored or reviewed drafts of the paper, and approved the final draft.

### Data Availability

The codes used for the experimentation and all details to compile and reproduce the experiments are available at GitHub: https://github.com/mawussi/ReSpaSol.

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
