# Peer review of "Performance impact of precision reduction in sparse linear systems solvers"

_PeerJ Computer Science, doi:10.7717/peerj-cs.778_

## Round 0.1 · original submission · Major Revisions

Based on reviewers’ comments, you may resubmit the revised manuscript for further consideration. Please consider the reviewers’ comments carefully and submit a list of responses to the comments along with the revised manuscript.

Reviewer 1 ·

Basic reporting

- The paper is clear and well-organized
- Sections are of appropriate sizes
- Intro. & Background are fine
- Figures are relevant

Experimental design

- The authors compared SpMV in single and double precisions on Intel CPU and NVidia GPU
- They also compared LU using PARDISO, cuSPARSE and SuperLU in single vs. double
- The fact that the authors are not performing a study for their own solver is also valuable to ensure a fair comparison

Validity of the findings

- The authors wrote that subnormal numbers can be an issue to achieve performance, and they illustrate with meaningful results
- They explained that such numbers can propagate during the computation and hurt the performance for numerous instructions
- They demonstrate that expecting a speedup of 2 just by using float instead of double is not possible in the majority of cases
- They pointed that incomplete factorization in float vs double has a lowest speedup compared to SpMV, which could motivate the implementation of a more efficient strategy

Additional comments

I think the paper deserves to be published as it provides interesting results that are beneficial for the community.
From my side, there are only a few points that should be addressed before having a final version.

- One thing that I had faced in the past is the poor implementation of single-precision kernels (in the MKL especially).
It might not be true anymore, but as the authors rely on the MKL/cuSPARSE as a "black-box", they are considering
that the kernels are as optimized in both cases, which might not be true.
I would like the authors to ensure that the differences they are seeing are not coming from a difference in the optimization level.
With this aim, I would suggest the following.
The authors could convert a small dense matrix that fit in the L1 cache in the CSR format and apply operations
(such as SpMV) on it (hundreds or thousands of times) such that it will provide a good view of the raw performance
of the kernels when the memory transfers are negligible.
The authors could then simply add a sentence in the manuscript to state if there is a difference or not (and
update the corresponding sections if needed).
This is just one way to do it, and of course, the authors could use any other strategy to test that the MKL
or cuSPARSE kernels are optimized similarly.

- I would appreciate in "EXPERIMENTAL SETUP" to have an idea of how the numbers were extracted,
for example, are they average of X executions? Or the median?

- Similarly, I imagine that the first data that will be used by the kernels are not in the L1 cache.
I would like the authors to confirm this point in "EXPERIMENTAL SETUP".

- I do not understand this sentence "because they are usually processed at the software level,"
My first assumption is to disagree with this statement. If I program in assembly a code
that does a dot product, it will work also with subnormal numbers without
the need to add anything "at the software level".
However, I think that instructions can have higher latency (need more clocks)
when subnormal numbers are used (or NaN as Intel in the past, etc.).
I would like the authors to clarify this point.

- I would suggest replacing "CSR" with "MXCSR" in the comments of Line 200 (to ensure no
ambiguity with CSR storage). I also suggest adding a number and a legend to this code
in order to describe what it does (without the need to go in the main text).

- I can imagine that the discussion concerning the bytes per values (12b/nnz ~ 8b/nnz)
is also valid for other operations than the SpMV.
I would consider that many of the symbolic/transformation stages are memory bound
and would not benefit from the shift from double to single (because it only reduces
the matrix memory by 1/3 and not 1/2).
But after a careful read, the authors made it clear by giving explanations
and stating that these stages are not related to computation
(so I think it will be clear for all readers).

- "rpecision." -> "precision."

·

Basic reporting

It would be better to add the definitions of "solve" in Figures 7 and 8, as well as Figures 9 and 10, because they seem to have different meanings.

Experimental design

This paper studies the effect of replacing double-precision arithmetic in solving sparse linear system with single-precision. The investigation about the mechanism how subnormal numbers are generated and caused poor speedup is interesting. Also, the finding of the lack of parallelism and floating-point arithmetic in the analysis and reordering phases of the solvers as the reasons of much lower speedup is reasonable.
However, it would be better if the authors could add some detailed analysis on the characteristics of matrices. For example, the matrix ss1 is small but shows low ratio of "Reordering & Analysis" ratio in Figure 7. This may caused the better speedup in Figure 1. So, the readers will be interested in why this matrix has shown different behavior.
Also, discussions on the results for iterative solvers are limited. It would be better to add the reasons why it has shown less speedup on GPU than CPU. In addition to that, there should be more description about the reason why the speedup of ILU is not significant.

Validity of the findings

No comment.

---

## Round 0.2 · accepted · Accept

Thank you for your submission to PeerJ Computer Science. Based on the reviewers' recommendation your article has been accepted for publication.

Reviewer 1 ·

Basic reporting

The authors carefully answer the questions/remarks, and I think the paper can now be accepted.

Again, the paper is clear and well-written.

Experimental design

The experimental part is rigorous and well done. Moreover, the code is publicly available.

Validity of the findings

The study of the effect of subnormal numbers, through the linear systems solvers, is important in the context of increasing use of mixed precision by the community.

Additional comments

None.

·

Basic reporting

No comment.

Experimental design

No comment.

Validity of the findings

No comment.